# Vertical Ramus Osteotomy, Is It Still a Valid Tool in Orthognathic Surgery?

**DOI:** 10.3390/ijerph191610171

**Published:** 2022-08-17

**Authors:** Oren Peleg, Reema Mahmoud, Amir Shuster, Shimrit Arbel, Shlomi Kleinman, Eitan Mijiritsky, Clariel Ianculovici

**Affiliations:** 1Department of Otolaryngology Head and Neck Surgery and Maxillofacial Surgery, Tel-Aviv Sourasky Medical Center, Sackler School of Medicine, Tel Aviv University, Tel Aviv 6423906, Israel; 2Maurice and Gabriela Goldschleger School of Dental Medicine, Tel Aviv University, Tel Aviv 6423906, Israel

**Keywords:** orthognathic surgery, vertical ramus osteotomy, sagittal split osteotomy, mandibular osteotomy

## Abstract

The purpose of this study is to evaluate mandibular osteotomy procedures during orthognathic surgery, with an emphasis on the complications of the two leading procedures: intraoral vertical ramus osteotomy (IVRO) and sagittal split osteotomy (SSO). We conducted a retrospective cohort study by extracting the records of patients who underwent either IVRO or SSO procedures during orthognathic surgery in a single center between January 2010 and December 2019. A total of 144 patients were included (median age of 20.5 years, 52 males). The IVRO:SSO ratio was 118:26 procedures. When referring to all surgeries performed, IVRO procedures were associated with shorter hospitalization than the SSO procedures, while the overall durations of surgery and follow-up periods were comparable. In contrast, when referring only to bimaxillary procedures, the duration of the IVRO bimaxillary procedures was significantly shorter than the SSO bimaxillary procedures. There were 53 complications altogether. Postoperative complications consisting of skeletal relapse, temporomandibular joint dysfunction, sensory impairment, and surgical-site infection were significantly fewer in the IVRO group. Both types of osteotomies have acceptable rates of complications. IVRO appears to be a safer, simpler, though less acceptable procedure in terms of patient compliance.

## 1. Introduction

A wide variety of mandibular osteotomy techniques in orthognathic surgery have been described [1], the most common of which is intraoral vertical ramus osteotomy (IVRO) and sagittal split osteotomy (SSO) [2]. Although SSO can be used for both mandibular prognathism and retrognathism deformities [3], IVRO is still widely used for the correction of mandibular prognathism [2]. 

IVRO procedures reportedly have a lower incidence of permanent neurosensory disturbance compared to SSO procedures [4]. Moreover, an IVRO procedure takes less time to perform [4], it is characterized by negligible cases of bad splits [5,6], and it is less harmful to the temporomandibular joint (TMJ) [7]. It has been reported that post-IVRO condylar changes could even induce a more favorable result in TMJ function [8]. However, the IVRO procedure may be inferior to SSO in some aspects. IVRO provides limited bony contact between the proximal and distal segments [6] and requires a period of postoperative maxilla–mandibular fixation (MMF) [9]. SSO may therefore be more suitable for cases where IVRO is not indicated [10].

Reports in the literature on skeletal stability are contradictory [11]. A meta-analysis by El Moraissi and Ellis [10] concluded that although SSO produced better vertical stability, the amount was clinically minor.

The purpose of this study is to evaluate mandibular osteotomy procedures during orthognathic surgery, with an emphasis on the complications of the two leading procedures: intraoral vertical ramus osteotomy (IVRO) and sagittal split osteotomy (SSO).

## 2. Materials and Methods

### 2.1. Study Design and Ethics

A retrospective cohort study was conducted according to the “Strengthening the Reporting of Observational Studies in Epidemiology” (STROBE) checklist [12]. Patients who underwent orthognathic surgery procedures in the Tel-Aviv Sourasky Medical Center (TASMC), Israel between January 2010 and December 2019 were identified according to the procedure codes of The International Classification of Diseases, 9th revision (ICD9) [13]. 

Surgical procedures were performed by one or two senior surgeons with more than 15 years of experience alongside an oral and maxillofacial resident. Only patients who underwent SSO or IVRO-based surgical procedures were included in the study.

Ethical approval was obtained on 21 June 2020 from the institutional Helsinki Ethics Committee (0300-20-TLV), which waived the requirement for informed consent.

### 2.2. Surgery

Bimaxillary procedures were carried out through a standard 1-piece Le Fort I osteotomy and one of the mandibular procedures (SSO based on Epker’s modification or IVRO). IVRO was used for a mandibular setback, while SSO was used only for mandibular advancement. In cases of bimaxillary procedures, the maxilla was operated on first.

MMF was part of the IVRO procedures, whereas in SSO procedures fixation was accomplished with a titanium 2 mm miniplate and screws. Patients were evaluated daily during their hospital stay. Patients were subsequently evaluated both clinically and radiographically during the follow-up outpatient visits.

### 2.3. Data Collection

Data were collected from the patient’s medical files and recorded onto structured Excel sheets. Independent variables included demographics, such as age and sex, type and duration of the surgical procedure, type, dosage and route of administration of prophylactic and postoperative antibiotics and steroids, hospitalization, the use of a urine catheter or a feeding tube, and the length of follow-up.

Dependent variables referred mostly to complications during the intraoperative or postoperative periods. Complications associated with maxillary procedures were excluded; thus, the complications collected were those related only to the mandibular osteotomies. Specifically, the intraoperative complications were

Hemorrhage: abnormal bleeding during surgery;Bad split: undesired intraoperative fracture of the mandible;Instrument failure: breakage of instruments, de-attachment of the orthodontic hardware into the surgical wound diagnosed during the procedure or by postoperative radiography.

Postoperative complications were subdivided into early (occurring during hospitalization up to 1 week post-surgery) and late complications (occurring after discharge from the hospital or at least 1 week postoperatively) defined according to the following diagnostic criteria:Hemorrhage: a bleeding event that requires treatment;Surgical-site infection (SSI) as defined by the Centers for Disease Control and Prevention [14];Infection other than SSI;Malocclusion: a postoperative abnormal occlusion requiring correction utilizing an orthodontic treatment;Skeletal relapse: a postoperative malocclusion that cannot be corrected by orthodontic treatment alone;Sensory impairment: any alterations in the sensory spread of the inferior alveolar nerve (IAN) branches evaluated using two-point discrimination, light touch, and pin-prick tests performed at each follow-up visit;Hardware complications: loosening, breakage, or failure of fixation plates and screws;TMJ disorders (TMDs) and muscular pain: pain, clicking, crepitation, and difficulty opening or closing the mouth;Systemic: significant changes in the patient’s baseline values, such as significant weight loss or a significant decrease in hemoglobin values.

Detailed information was collected on the onset, duration, management, and outcome of each complication sustained by each patient.

### 2.4. Statistical Methods

Categorical variables were reported as numbers and percentages. The Shapiro–Wilk test was used to evaluate the normal distribution of continuous variables, which were presented as the median and interquartile range (IQR). The Mann–Whitney test was used to compare continuous variables between the two types of surgeries. Fisher’s exact test and the Chi-square test were applied to compare categorical variables. The propensity score was calculated as the probability of patients undergoing a bilateral SSO. Logistic regression was used to calculate the propensity score. Age, sex, number of operated jaws, and performance of genioplasty were included in the propensity score. Multivariable logistic regression was used to study the association between the type of procedure and the studied outcomes. All statistical tests were 2-sided. A *p*-value < 0.05 was considered statistically significant. NCSS 2020 software was used for all statistical analyses (“NCSS 2020 Statistical Software (2020). NCSS, LLC. Kaysville, UT, USA, ncss.com/software/ncss (accessed on 7 September 2020)”).

## 3. Results

### 3.1. Participants

A total of 473 medical files were extracted according to the designated codes, of which 138 were excluded due to duplication and 98 were excluded due to the patients having undergone a surgical procedure other than orthognathic surgery. Of the remaining 237, 67 were surgeries involving only the maxilla and thus were excluded. The remaining 170 patients had undergone either single-jaw mandible or bimaxillary-based-SSO or IVRO procedures. A lack of clinical information led to the exclusion of 26 of those patients. A total of 144 patients comprised the final study population.

### 3.2. Descriptive Data

A total of 52 males (36.11%) and 92 females (63.89%) had undergone IVRO and SSO procedures between January 2010 and December 2019 in TASMC. Their median age was 20.5 years (IQR 18–24 years). Most patients had undergone bimaxillary procedures (133/144, 92.36%). The majority of these surgical procedures were IVRO (*n* = 118, 81.94%), and the rest (*n* = 26, 18.06%) were SSO. The differences between the overall duration of surgery and the length of the follow-up period were not significant between these two groups (Table 1).

The IVRO and SSO operation times for the single-jaw procedure group (median 2.075 h, IQR (1.45–2.7) and median 2.7 h, IQR (2.48–3.48), respectively) were not significantly different (*p* = 0.281), whereas the IVRO operation time (median 4.27 h, IQR (3.99–4.8)) and the SSO operation time (median 5.5 h, IQR (5.02–6.5)) for the bimaxillary procedure group were significantly different (*p* < 0.001).

The two groups were statistically comparable concerning intraoperative and early postoperative complication rates. Late postoperative complications, however, were significantly different between the groups (Table 2).

### 3.3. Main Results

A total of 53 complications were registered in our total study group. The majority of these complications were sustained by the patients in the bimaxillary group who underwent an IVRO procedure (*n* = 38). The distribution of these events is listed in Table 3.

#### 3.3.1. Single-Jaw IVRO vs. Single-Jaw SSO Complications

No complications were recorded for the patients in the single-jaw IVRO group. Complications in the single-jaw SSO group consisted of one intraoperative case of a bad split. In the late postoperative period, one case of skeletal relapse, one case of TMD, and one case of sensory impairment occurred. In addition, three SSIs required drainage. One occurred during hospitalization and was drained bedside, and two occurred during the late postoperative period and required drainage under general anesthesia. There was no significant difference in the complication rate between these two subgroups (*p* = 0.49).

#### 3.3.2. Bimaxillary LeFort I-IVRO vs. Bimaxillary LeFort I-SSO Complications

There were eight complications in the LeFort I-SSO group compared to 38 complications in the LeFort I–IVRO group. Intraoperatively, there was one bad split and one instrument breakage in the IVRO group, and no complication in the SSO group. Within 1 week from surgery, the number of complications in the SSO group rose to two, and both were systemic. In contrast, there was a clear diversity in the IVRO group: systemic complications and malocclusions accounted for four cases each, non-SSIs accounted for two cases, and one patient underwent draining twice due to an SSI.

Finally, in the late postoperative period, six complications had been reported in the SSO group. They consisted of two cases of sensory impairment that were treated with light laser therapy, and one case of malocclusion, skeletal relapse, fixation hardware complications, and nonsurgical site infections. In the IVRO group, sensory impairment accounted for six cases, while TMJ muscular-related pain accounted for five cases. One patient required TMJ arthrocentesis, another patient had partial excision of his left masseter muscle due to masseteric hypertrophy, and three other patients were treated with light laser therapy due to IAN sensory damage. Malocclusion and SSIs accounted for four cases each. All SSI cases required drainage under general anesthesia. Systemic complications and skeletal relapse accounted for two cases each, and there was also one case of nonsurgical site infection. Another patient had a chin deviation and was categorized as “other.” There was no significant difference in the rate of complications between these two subgroups (*p* = 0.322).

## 4. Discussion

SSO and IVRO are used worldwide to treat mandibular deformities during orthognathic surgery [2]. Although the SSO procedure is more commonly used in orthognathic surgery around the world, in our institution, it is customary to use IVRO to execute mandibular setbacks and SSO for mandibular advancement. Since most of our patients present with skeletal class III deformities, more than 80% of the mandibular surgeries include IVRO. The purpose of the current study was to compare complication rates of the two most common mandibular osteotomy procedures during orthognathic surgery (IVRO or SSO) while trying to determine whether the IVRO procedure still holds a valid place in the surgeon’s toolbox.

When surgical procedures involving both jaws were compared, a LeFort I osteotomy + SSO took an average of more than 1 h longer than a LeFort I osteotomy + IVRO. Similar results were reported in the literature [5]. Postoperative hospitalization was also significantly longer among the SSO patients.

As opposed to reports in the literature that describe unexpected fractures being more common in SSO procedures in which there is a greater effect on bone architecture and density [15], and the presence of a third molar [16], we did not find any significant differences in intraoperative complication between the two procedures. In contrast, the main differences in postoperative complications between the two procedures, which were also found to be significant, were related to those that occurred in the late postoperative period.

The SSO procedure is more prone to neural injuries than the IVRO procedure since the IAN can be damaged during the sagittal osteotomy, the separation of the proximal and distal segments, and the fixation [15], while it is only exposed to damage during osteotomy during the IVRO procedure. The literature describes a temporary neural injury rate higher than 90% during SSO procedures [10], compared to a rate of 35% during IVRO procedures [17]. In our study, 11.54% of IAN injuries occurred in the SSO group, compared to 5.08% in the IVRO group.

One of the benefits of the IVRO procedure is the ability to reposition the condyle so that it is stress-free and to alter the disc–condyle relationship when indicated, in contrast to the SSO procedure where fixation of the proximal and distal segments with plates and screws may alter the condyle position and possibly cause functional instability and relapse [8]. Although the incidence of internal derangement of the TMJ is low after SSO, it is even lower after IVRO [18]. In our study, the TMD rates were 3.85% in the SSO group and 4.23% in the IVRO group. Since we do not have any information about the condition of the patients’ TMJs before the surgery, we cannot determine whether the surgeries we performed worsened or improved the patients’ symptoms.

The stability of any mandibular ramus osteotomy is affected by the amount of retrusion/protrusion, the rigidity of fixation, and the continuous growth of the mandible [2]. Skeletal relapse following SSO procedures is usually correlated to an anterior open bite development. Our study findings showed higher incidences of skeletal relapse after SSO (7.69%) compared to IVRO (1.69%).

Fixation hardware failure is related only to the SSO group. The common reasons for its occurrence include pain, discomfort, exposure to the hardware, and loosening of screws causing infection [19]. Fixation hardware failure in SSO reportedly ranges between 7% and 16% [19]. We had only a single case of hardware failure (3.84%). It should be borne in mind that plate removal does not mean treatment failure since the plates were removed after a healing period that allowed bony union.

Although the percentages of non-SSI and systemic complications were higher in the SSO group, we assume that they are less related to the type of osteotomy and more related to the fact that surgical intervention had been carried out. We believe that the reason lies in the patient being more exposed to infections since the SSO procedure takes longer to perform than the IVRO procedure and involves longer hospitalization.

The major disadvantages of the IVRO procedure warrant some emphasis. IVRO dictates the use of MMF for 6 weeks, which patients tend to oppose since it creates considerable feeding and speech-related inconvenience [20]. In our experience, a proper explanation of the pros and cons of each procedure, especially the less risk of nerve damage in IVRO, usually mollifies their concerns.

Virtual surgical planning was not addressed in our article. In a recent meta-analysis comparing the effectiveness of traditional (TSP) and virtual surgical planning (VSP) for orthognathic surgery, VSP and patient-specific osteosynthesis were found to be significantly better at predicting certain reference areas, in addition to the advantage of reducing the surgical times, even for inexperienced surgeons. Patient-specific cutting guides and osteosynthesis mean less risk for intraoperative bad splits and neural injuries, while a shorter operative time means less blood loss, less anesthesia time, and lower surgical risk [21]. 

Within the limitations of this study and the use of traditional surgical planning, IVRO and SSO have acceptable rates of complications. However, skeletal stability in IVRO-related procedures was superior to that of SSO-related procedures, and sensorineural damage occurred more with SSO than with IVRO procedures. We believe that the IVRO procedure can be executed safely, with fewer complications and therefore should not be abandoned when considering mandibular setbacks. Nevertheless, a more comprehensive comparison between the two procedures, during mandibular setbacks and while addressing the effect of virtual surgical planning, is needed.

## 5. Conclusions

In the hands of experienced surgeons, both types of mandibular osteotomies, IVRO and SSO, provide the desired results and present acceptable rates of complications, where many of these complications are reversible. IVRO appears to have some advantage over SSO in this matter, though it carries less patient compliance.

## Figures and Tables

**Table 1 ijerph-19-10171-t001:** Characteristics of the study groups.

Characteristic	IVRO	SSO	*p*-Value
Age (years)			0.017
Median	20	24	
IQR	18–23	18–29	
Operation duration (hours)			0.281
Median	4.26	4.88	
IQR	3.98–4.79	2.88–6.06	
Hospitalization (days)			0.032
Median	7	7	
IQR	5–7	6.75–8	
Follow-up (weeks)			0.136
Median	7.14	17.5	
IQR	4.43–20.14	6.36–45.29	

IQR, interquartile range; IVRO, intraoral vertical ramus osteotomy; SSO, sagittal split osteotomy.

**Table 2 ijerph-19-10171-t002:** Demographic and clinical data of the study groups.

Characteristic	IVRO	SSO	*p*-Value
*n* (%)	118 (81.94)	26 (18.06)	
Sex			0.047
Male	47	5	
Female	71	21	
Surgical procedures			<0.001
Single jaw	2	9	
Bimaxillary	116	17	
Feeding tube use	90	17	0.386
Catheter use	99	17	0.04
Complications			
Intraoperative	2	1	0.452
Early postoperative	11	3	0.718
Late postoperative	25	11	0.048

IVRO, intraoral vertical ramus osteotomy; SSO, sagittal split osteotomy.

**Table 3 ijerph-19-10171-t003:** Distribution of complications in each group.

Complication	Single Jaw IVRO	Single Jaw SSO	Bimaxillary (IVRO + LeFort I)	Bimaxillary (SSO + LeFort I)
Intraoperative Bad split	-	1	1	-
Instruments’ breakage	-	-	1	-
Early postoperative				
Malocclusion	-	-	4	-
Surgical site infection	-	1	1	-
Non-surgical site infection	-	-	2	-
Systemic	-	-	4	2
Late postoperative				
Malocclusion	-	-	4	1
Skeletal relapse	-	1	2	1
TMJ-related	-	1	5	-
Fixation hardware	-	-	-	1
Neural damage	-	1	6	2
Surgical site infection	-	2	4	-
Non-surgical site infection	-	-	1	1
Systemic complication	-	-	2	-
Other complications	-	-	1	-

IVRO, intraoral vertical ramus osteotomy; SSO, sagittal split osteotomy; TMJ, temporomandibular joint.

## Data Availability

The data that support the findings of this study are available on request from the corresponding author. The data are not publicly available due to ethical and legal restrictions.

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
