# Peer review of "Vertical Ramus Osteotomy, Is It Still a Valid Tool in Orthognathic Surgery?"

_ijerph, 2022, doi:10.3390/ijerph191610171_

Round 1

Reviewer 1 Report

"Regarding digital planning, all the data presented here were carried out in the traditional way and not the digital approach, therefore, the digital aspect was not addressed in the article. The digital planning came into use in our department only in the last two months and we still have no data to present or to discuss. There is no doubt that in the future it will be interesting to compare the cases presented here with the cases carried out in with the help of digital planning. Following your comment we included in the discussion part and to the reference list additional studies dealing with the topic that you suggested."

The comparison with the literature inherent to the topic can be further discussed and deepened.

Best regards

Author Response

We thank the Reviewer for the valuable comment and we agree with it. There is no doubt that the field of Orthognathic surgeries is currently aimed at computerized planning, something that requires our addressing in the article. A paragraph dealing with virtual planning was added according to this comment and our conclusions have been edited and revised based on this.

Reviewer 2 Report

the manuscript can be published in the present form

Author Response

Thank you for your previous comments. 

They have definitely contributed to the improvement of the article.

This manuscript is a resubmission of an earlier submission. The following is a list of the peer review reports and author responses from that submission.

Round 1

Reviewer 1 Report

The manuscript is written in a fluent way.
I suggest reviewing some typos and spelling errors.
Pay more attention to the description of the results.
Revise the discussion, introducing a greater comparison with studies already present in the literature on the same topic, also in the light of progress in terms of orthognathic digital programming in terms of predictability of the result and management / reduction of complications.

Best regards

Reviewer 2 Report

The authors do not specify that IVRO must not rigidly fixed therefore requires a period of intermaxillary fixation of at least 21 days if bilateral ad 15 days if monolateral which is certainly a big discomfort for the patients and a potential cause of reduction in the patients' maximum opening capacity. Furthermore, they do not mention in the complications of IVRO the possibility of having a pseudarthrosis due to a smaller bone contact surface and the possibility of having a double occlusal position with a discrepancy between centric relationship and individual dental occlusion is much greater in IVRO compared to SSO. For everything mentioned here, the conclusion that the "VRO appears to be a safer, simpler, though less acceptable procedure in terms of patient compliance" is not accetable. The bibliography is certainly old with 12 out of 17 articles dating back 10 years or more. In the end, the significant number of complications when compared to the total number of SSO seems more due to a lack of experience of the authors, as they have declared, rather than to the problem of a technique so well established over the years.